# Lethal Concentration and Sporulation by Contact and Direct Spray of the Entomopathogenic Fungus *Beauveria bassiana* on Different Stages of *Nezara viridula* (Heteroptera: Pentatomidae)

**DOI:** 10.3390/jof8111164

**Published:** 2022-11-04

**Authors:** Maribel Portilla, Minling Zhang, James Paul Glover, Gadi V. P. Reddy, Chris Johnson

**Affiliations:** USDA-ARS Southern Insect Management Research Unit, 141 Experiment Station Rd., P.O. Box 346, Stoneville, MS 38776, USA

**Keywords:** southern green stink bug, entomopathogenic fungi, lethal concentration 50, *Beauveria bassiana*, agricultural pest

## Abstract

The southern green stink bug, *Nezara viridula* (L.) (Heteroptera: Pentatomidae) is the most significant pest of soybean worldwide. The present study was conducted to compare the effectiveness of a Delta native strain NI8 of *Beauveria bassiana* by contact and direct spray on nymphs (2nd to 5th instar) and adults of *N. viridula*. Water control and four concentrations of *B. bassiana* were used to evaluate the survival, mortality, and molting percentage and to estimate median lethal concentration (LC_50_), median lethal sporulation (LS_50_), and resistance ratio (RR_50_). Direct spray at all concentrations observed the greatest reduction in survival on all life stages. Mortality and sporulation were positively correlated by concentration, while molting was highly variable with a significantly lower negative correlation on insects that were directly sprayed. Pathogenicity exhibited reduction as young stages developed and emerged to adult. The LC_50_ (Contact: 612 spores/mm^2^; Direct spray: 179 spores/mm^2^) and LS_50_ (Contact: 1960 spores/mm^2^ Spray: 3.3 × 10^6^) values showed that adults of *N. viridula* were highly resistant than any other life stage when exposed to either contact or direct spray. Fourth instar was the most susceptible (LC_50_: Contact: 18 spores/mm^2^; Direct spray: 23 spores/mm^2^) (LS_50_: Contact: 53 spores/mm^2^; Direct spray: 26 spores/mm^2^) followed by second, third, and fifth instars.

## 1. Introduction

The southern green stink bug, *Nezara viridula* (L.) (Heteroptera: Pentatomidae), is an herbivore invasive insect responsible for reducing production in soybean, *Glycine max* (L.) (Fabales: Fabaceae) and cotton, *Gossypium hirsutum* (L.) (Malvales: Malvacea), both quality and quantity [1,2,3]. In addition to the plant pathogens transmission, adults and nymphs can injure those crops at any phenological growth stage [2,3,4,5]. Globally, the low susceptibility of *N. viridula* to pyrethroids and organophosphates is well known, yet, chemical control remains the primary tool to manage this pest in most of southern states of U.S., and tropical and subtropical zones in five continents [1,6,7,8,9,10,11,12,13,14].

Although, attempts with biological control for *N. viridula* dated from the early 19th century [15,16], they are still in progress worldwide because of the lack of success due to their low efficacy [16]. Its classical biological control mainly involved egg parasitoids and native parasitoids of adults [17,18,19] only; no attempts has been done using entomopathogenic fungi, although several studies have demonstrated the effectiveness of *Beauveria bassiana* (Balsamo-Crivelli) Vuillemin (Ascomycota, Hypocreales: Cordycipitaceae) against *N. viridula* [7,20,21,22,23,24,25,26,27]. Somehow, it is understandable why the lack of interest to control *N. viridula* with pathogenic fungus. The efficacy of *B. bassiana* for insect control is often influenced by environmental conditions that could affect positively under laboratory and limited efficacy in the field [28]. Uncontrol temperature, UV-light, and humidity, are the main factors that can be attributed for an unsatisfactory control [29,30,31]. However, factor of insect behavior like grooming [32], thermoregulation [30], and molting [33] can be also added to the contribution of negative effects of insect control. None of these factors can be controlled under field conditions, making it difficult to present pathogenic fungi to growers as a potential alternative for pest control. Among all those uncontrollable conditions, the integument composition of lipids and phenols of *N. viridula* adults that limit the attachment to the insect integument [34,35,36,37,38] could be added to the problem for using *B. bassiana* as one of the control agents alone to suppress *N. viridula* population. Therefore, it is important to understand that no pest management tactic acts independently, it must be part of an integrated pest management (IPM) program, where microbiological agents can be adopted in consideration of insect management and vice versa [39].

The adverse environmental effects of chemical insecticides have increased the interest in microbiological agents over the years. Regardless of the pathogen’s slow action, *B. bassiana* is well recognized as an essential source of myco-pesticides used for numerous insect pest all over the word [40]. *Beauveria bassiana*, is one of the most common pathogens that occur naturally in North America that could be contributed to building a sustainable environment [41]. In the Mississippi delta, natural infection of endemic *B. bassiana* strains has been noted in several species of sucking insects including *Lygus lineolaris* (Palison de Beauvois) (Hemiptera: Miridae) [42], *Piezodorus guildinii* (Westwood), *Chinavia hilaris* (Say) (Hemiptera: Pentatomidae), and nymphs and adults of *N. viridula* [Portilla et al. unpublished data]. Therefore, this study was conducted to evaluate the effectiveness of a Delta native *B. bassiana* strain NI8 on different instars and adults of *N. viridula*. While several studies on the effect of *B. bassiana* on *N. viridula* adults are available, still there is lack of information on the lethality of its different instars, and no information on methods of infectivity has been reported. This investigation evaluated two infection methods by contact and direct spray and estimated lethal mortality, lethal sporulation, and resistance ratio of 2nd, 3rd, 4th, 5th instars, and adults of *N. viridula*. Correlation on mortality, sporulation, and molting percentage among concentrations and method of infection was also calculated.

## 2. Materials and Methods

### 2.1. Insect Colonies

*Nezara viridula* colonies have been initially collected in 2019 from several counties in Mississippi Delta and maintained on an artificial diet for successive generations at the United States Department of Agriculture (USDA), Agricultural Research Service (ARS), Insect Management Research Unit, Stoneville, MS, US. Insects were kept following the method described in [43], which was developed to produce large numbers of even-ages nymphs and/or adults. Insects were held in environmental incubators with 16: 8 h Light: Dark (L:D) photoperiod, 27 °C, and 65% Relative Humidity (RH). Mixed sex adults 1–2 d old and second to fifth instar nymphs 1 d after molting were used for bioassays. Three colonies were selected from the insect rearing, one colony per repetition.

### 2.2. Fungal Isolate

Stock conidia suspensions of *B. bassiana* Delta native NI8 strain was obtained from the USDA-ARS, Southern Insect Management Research Unit (SIMRU), Stoneville, MS, US. Each conidia suspension serially diluted and suspended in 50 mL of 0.04% Tween-80 (Sigma-Aldrich P8074, Darmstadt, Germany) came labeled as follows: 95% spore germination and 5 × 10^4^ (380 spores/mm^2^), 5 × 10^5^ (38 spores/mm^2^), 7 × 10^6^ (3.8 spores/mm^2^), 7 × 10^7^ (0.38 spores/mm^2^). The stock conidia suspensions were kept in an incubator at −4 °C (Percival Scientific I-36LL, Perry, IA, USA).

### 2.3. Bioassay Procedures

#### 2.3.1. Infectivity by Contact: Treated Diet

A non-autoclaved solid diet was used for this experiment [44]. The diet consisted of: Agar 70 g (www.eapcolloids.com, accessed on 2 June 2022), toasted wheat germ 100 g (Nutritional Design Inc. 510-16, Lynbrook, NY, USA), coarsely ground lima bean meal 150 g (Bio-Serv G1305, Flemington, NJ, USA), toasted soy flour 25 g (Processor’s Choice 063, Moody, AL, USA), sugar 16.5 g (local grocery store), benzoic acid 3 g (Sigma-Aldrich B9300-500), propionic acid 4 mL (Sigma-Aldrich PI386), phosphoric acid 0.5 mL (Sigma-Aldrich 696017-2), methylparaben 3.8 g (MP Biomedical 10234, mpbio.com, accessed on 10 June 2022), sorbic acid 3.8 g (MP Biomedical 102937), and aureomycin (Sigma-Aldrich). All weighed components were mixed and blended with 3000 mL of boiling water and 12 egg yolks (local grocery store) for about 4 min in an industrial blender Waring-CB15In (www.katon.com, accessed on 15 April 2022). Five mL of the final mix were poured into individual plastic cups (T-125 Solo Cup Company, www.solocup.com, accessed on 10 March 2022) and kept at room temperature to cool and solidify before use. A total of 1500 diet cups were prepared for this experiment (750 cups for infectivity by contact and 750 cups for infectivity by direct spray).

Aliquots of 6 mL of each treatment suspension and water control were sprayed to each repetition (three repetitions/treatment) (10 diet cups/rep/treatment). Each repetition was sprayed with a different fungal suspension. Ten diet cups (experimental unit) were placed in a standard test sieve (24 cm No. 20 stainless steel sieves ENDECOTTS) (mesh prevented accumulation of spore solution) and located in the middle of a 38.5 cm diameter area inside the spray tower equipped with a spraying system Co. FN5925-001-001A. Treatments were applied from lowest to highest concentration to prevent cross-contamination. After each application, the sprayed replication (10 diet cups) was left on the counter for 10 min to let them dry before exposing them to the insects. Insects for each repetition were selected from the assigned colonies (three colonies) and released individually per sprayed diet cup (2nd, 3rd, 4th, 5th instars, and adults). The selection of each instar was done by following the identification described by [43] as follows: second instar: body almost entirely black about 2–3 mm in length; third instar: black body with white spots about 4 mm in length; fourth instar: black body with white and rose spots, greenish thorax, and head, about 6 mm in length; fifth instar: light form: Same description as the fourth instar, about 10 mm in length with wing pad marks; dark form: same description as light form but nearly black head, thorax, and abdomen. Exposed nymphs and adults to the treated diet were examined daily for molting and mortality rates. Dead insects were kept individually in the same cup and continued a daily observation for the presence absence of sporulation. Insects were held for 10 d in an environmental chamber at 27 °C, 65% Relative humidity, and photoperiod of 12:12 (L:D) h.

#### 2.3.2. Infectivity by Direct Spray

Similarly, as it was for infectivity by contact, aliquots of 6 mL of each treatment suspension and water control were sprayed to each repetition (3 repetitions/treatment) (10 individuals/repetition/treatment) (2nd, 3rd, 4th, 5th instars, and adults). Each repetition was selected from the assigned insect colonies (three colonies) and sprayed with a different fungal suspension. Ten selected insects by the specific stage (experimental unit) were placed in a plastic container (15 cm in diameter) with the base covered with a filter paper (filter paper allowed insects walked and prevented accumulation of conidia solution) and sprayed following the process previously described for infectivity by contact. Nozzle was changed for this experiment. After each spray (10 nymphs or adults/replica), (experimental unit) nymphs or adults were released in insect rearing cages (30.5 mm 3; Bio-Quip 1466A, Los Angeles; CA, USA) (one cage per replication) (15 cages total) and let them there for 10 min until insect dried. The sprayed insects were then placed individually in diet cups that were previously prepared. Insects were examined daily for molting and mortality rates. Dead insects were kept individually in the same cup and continued a daily observation for the presence or absence of sporulation. Insects were held for 10 d in an environmental chamber at 27 °C, 65% Relative humidity and a photoperiod of 12:12 (L:D) h.

### 2.4. Statistical Analysis

Molting, mortality, and sporulation were analyzed by using SAS 9.4, a one-way Analysis of Variance (ANOVA) followed by Tukey’s HSD test [45]. A factorial randomized complete block design with factorial arrangements (concentration x replicates) was used for both infectivity by contact and direct spray. Lethal concentration, lethal sporulation, and resistance ratio were analyzed with PROC PROBIT-SAS procedure using log base 10 of the concentrations. Mortality for each bioassay was corrected for control effects using Abbot’s formula [46]. Confidence intervals and RR_50_ were calculated using the method of Robertson and Priestler [47]. To determine correlation between variables (molting, mortality, and sporulation) and concentration, linear and cubic regressions were calculated using SIGMA-PLOT 14.0. Nonparametric estimates of survival were compared among treatments-concentrations using PROC LIFETEST-SAS.

## 3. Results

### 3.1. Survival Probability

#### 3.1.1. *Nezara viridula* Nymphal Stage

Both methods of infectivity demonstrated that *N. viridula* was susceptible to *B. bassiana*. Significant differences in survival were observed in all instars among concentration regardless of the method of infectivity, except for 4th instar-direct spray (*p* = 0.621) (Table 1). Figure 1 and Figure 2 show that mortality occurred early in insects exposed to higher concentrations and a day earlier in 3rd and 4th instar treated by direct spray. Based on the Product-limit survival estimate (LIFETEST), higher concentration of *B. bassiana* could kill 2nd instar by contact or 5th instar by contact and/or direct spray 24 h post-exposure (Figure 1A–D and Figure 2A–D). Mortality in the water control will not occur until 6–7 d post water treatment. Survival decreased gradually and reached the maximum mortality 10 d after treatment, mainly for the highest concentration.

#### 3.1.2. *Nezara viridula* Adults

The survival probability test indicated that, similarly to the fifth instar, adults also can be killed either by contact or direct spray 24 h after spray only with the highest concentration (7 × 10^7^) (Figure 3). Mortality with the lowest concentration was observed at day 4 and day 5 after exposure to contact and direct spray, respectively. At the same time, control was recorded with the latest mortality that started on days 7–8 after water treatment (Figure 3A,B). The survival curves demonstrated that for adults the infectivity by direct spray is more effective than contact at all concentrations. Although survival had an interaction, no significant differences were found among concentrations by direct spray (*p* = 0.0692) (Table 1).

### 3.2. Mortality, and Sporulation Percentage of Nezara viridula Nymphs and Adults among Concentration of Beauveria bassiana Strain NI8

Differences in mortality and sporulation percentage on both method of infection are described via cubic regression. Figure 4 shows a positive correlation between *B. bassiana* concentrations and mortality and sporulation rates with contact and direct spray. Although, there were no significant differences between the 3rd and 4th instars between infectivity methods (Figure 4B,C), both mortality and sporulation percentages were consistently greater in the direct spray bioassays at all test concentrations for all life stages than the contact bioassay (Figure 4A–J). The highest mortality 10 d post exposure was found for 2nd and 3rd instar (100%) with the highest concentration (7 × 10^7^) by direct spray (Figure 4A,B). The pathogenicity decreased as the nymphs developed to the late instars and emerged as adult, where the mortality for 4th instar was 96.7 ± 2.0% (SD) (Figure 4C), 86.7 ± 3.5% (SD) for 5th instar (Figure 4D) and adults decreased to 56.7 ± 5.1% (SD) (Figure 4D) as it was expected.

Highly significant differences in life stages mortality among concentration were found by direct spray (df = 4, 149, *p* ≤ 0.0001) for 2nd instar: *F* = 26.01, 3rd instar: *F* = 22.20, 4th instar: *F* = 18.04, 5th instar: *F* = 13.63, and adult: *F* = 6.25 and by contact (df = 4, 149, *p* ≤ 0.0001) for 2nd instar: *F* = 16.06, 3rd instar: *F* = 9.95, 4th instar: *F* = 21.23, 5th instar: *F* = 6.29, and adult: *F* = 3.78 with an obvious decreasing pathogenicity pattern from younger to older instars to adult stage at all concentrations. Sporulation never was equal to mortality for any life stage at any concentration. However, it was always greater on nymphs that were directly sprayed (Figure 4F–I) except for 3rd and 4th instars that no significant differences were found with the highest concentration (7 × 10^7^) by contact: 83.0 ± 3.5% (SD), 82.0 ± 3.8% (SD) and direct spray: 77.0 ± 3.1% (SD), 76.0 ± 3.8% (SD), respectively (Figure 4G,H). In contrast to instar nymphal, sporulation for adults was slightly higher on insects that were exposed by contact to infected diet at all concentrations (Figure 1J) with 3.3 ± 1.8, 3.3 ± 1.8, 10.0 ± 3.1, and 26.7 ± 4.5% (SD) at 5 × 10^4^, 5 × 10^5^, 7 × 10^6^, and 7 × 10^7^, respectively. No sporulation was found in water control, neither by contact nor direct spray.

### 3.3. Lethal Mortality, and Sporulation Response of Nezara viridula Nymphs and Adults to Beauveria bassiana Strain NI8

Both infection methods showed that LC_50_s and LS_50_s as determined by the PROC PROBIT TEST, indicated variability among life stages of *N. viridula*. Table 2 and Table 3 showed that younger nymphs were more susceptible than the oldest instar and adults. There were not significant differences in susceptibility between younger nymphs 10 d after exposure to either method of infectivity. However, the LC_50_ by direct spray was 1.9-fold lower for 2nd instar and 2.8-fold lower for 3rd instar than those nymphs exposed by contact. Unexpectedly, a lower LC_50_ was obtained for 4th instars when they were exposed by contact with 17.8 spores/mm^2^ (Table 2), which was 1.3-fold lower than direct spray (22.8 spores/mm^2^) (Table 3). As it was noted before, the susceptibility decreased as the nymphs reached to 5th instars, increasing the LC_50_ to 10.2-fold (139.6 spores/mm^2^) for 5th instars and to 11.4-fold (179.4 spores/mm^2^) for adults. That lethal mortality was even greater when 5th instar (159.6 spores/mm^2^) and adults (612 spores/mm^2^) were exposed to *B. bassiana* by contact.

The pattern for LS_50_s varied for young nymphs compared to LC_50_s. There were no significant differences between 3rd and 4th instars regardless method of infection (Table 4 and Table 5). Second instars needed higher concertation to sporulate by direct spray (78.2 spores/mm^2^) and 2.5-fold higher (197.3 spores/mm^2^) when the infection was by contact. The resistance increased 16.4-fold for 5th instars (869 spores/mm^2^) and 37-fold for adults (1960 spores/mm^2^) when exposed to *B. bassiana* by contact (Table 4). Table 5 shows that LS_50_ value was lower for 5th instars by direct spray (563 spores/mm^2^), but unexpectedly much higher for adults (3.3 × 10^6^) when compared with contact.

### 3.4. Molting Percentage of Nezara viridula Nymphs among Concentrations of Beauveria bassiana Strain NI8

Differences in molting percentage on both methods of infection are described via linear regression. Figure 5 shows high variability in molting percentage among concentration mainly on nymphs exposed to *B. bassiana* by contact. There was a moderate negative correlation between *B. bassiana* concentrations and molting rates for 2nd (r^2^ = 0.93) and 3rd instars (r^2^ = 0.62) infected by direct spray (Figure 5A,C). No exuviae were found with sporulation on any nymphal stages at any concentration, neither by contact nor by direct spray.

## 4. Discussion

Currently, in U.S and in all soybeans producing countries where *N. viridula* is causing economic damages; its control continues relying on the use of heavy insecticides applications; due to all non-chemical control methods developed since more than a century ago appear to be inadequate for the control of *N. viridula* (sterile insect technique, inundative and conservation biological control, and trap cropping) [48]. All these non-chemical approaches have been incorporated into IMP programs except for biopesticides because stinkbugs are naturally resistant to fungal infection since the aldehydes, that are part of their defense secretions, serve as antimycotic agents against certain entomopathogenic fungi like *B. bassiana* [38,49]. Yet, efforts continue to better understand the fungistatic activities with *Nezara*’s cuticle. In general, our results corroborated, the high tolerance that adults have to *B. bassiana* that only elevate concentration of this entomopathogenic fungus will increase the probability to kill this insect. In our study the LC_50_ values of 1.1 × 10^7^ spores/mL and 3.2 × 10^7^ spores/mL estimated for dead adults by contact and direct spray, respectively, are comparable to [38] who reported 286 × 10^7^ spores/mL under laboratory condition or [1] with estimated values of LC_50_ of 19.6 × 10^7^ spores/mL under field conditions.

No study was found with separated estimated values of mortality and sporulation; therefore, based on our result, much greater concentrations (3.6 × 10^8^ spores/mL by contact and 5.5 × 10^11^ spores/mL by direct spray) will be required to complete the infection. Sporulation percentage never was equal to mortality; however, sporulation percentages were closer to nymphs’ mortality that it was for adult mortality. Similar results were found for nymphs by [50] that obtained 93.7% mortality and 60.5% sporulation for 3rd instars, 87.5% mortality, 68.7% sporulation for 4th instars, and 100% mortality and 87.5% sporulation for 5th instars. However, these results differed for adults, where higher mortality (93.7%) and sporulation (75%) were obtained compared with our study that even with the highest concentration (7 × 10^7^) the mortality did not reach more than 60% mortality and 25% sporulation by contact or direct spray. It has been demonstrated that the attachment of the conidia to the *N. viridula* adult cuticle does not exceed 2% [38]; but the attached conidia will germinate between 18 or 48 h [24], which seems to be more than enough to kill adults of *N. viridula* if the environmental conditions are favorable for the fungus reproduction.

The high susceptibility of younger nymphs it was demonstrated in this investigation, which corroborates with other studies [27,50,51,52,53]. In the current study, the tolerance increased 9.3-fold by contact and 6.1-fold by direct spray when 4th instars developed to 5th instars. The LC_50_ values for 2nd (36 spores/mm^2^, 19 spores, mm^2^) 3rd (44 spores/mm^2^, 16 spores/mm^2^), and 4th (18 spores/mm^2^, 23 spores/mm^2^) instars by contact and direct spray, respectively, exhibited that the molting process that occurred as part of their development did not interfere with the fungus infection. This could explain the high molting percentage variability observed in our study with no correlation among concentration when instars were exposed by contact, and a moderate negative correlation for 2nd and 3rd instars, when they were exposed to the direct spray. No exuvia was found with sporulation for any instars at any concentration, neither by contact nor direct spray.

The two methods of infection used in this study indicated, that spraying insect directly could be more efficient in controlling nymphs or adults of *N. viridula*; however, the infection obtained by contact give a better understanding that both nymphs and adults are capable to pick up spores from infected surfaces. Similar observation was found by [49], who noted that the antennal tips, tarsi, and distal region of tibia of *N. viridula* both nymphs and adults, were high-affinity areas for spore adhesion and/or entrapment. In general, based on our results and in many other studies [1,7,13,14,19,27,38,50,51,52,53] *B. bassiana* could be considered a potential non-chemical approach to be adopted for use in integrated management of this pest. The adult’s high tolerance to *B. bassiana* cannot be a limitation, considering that all its life stages, including eggs, are highly susceptible to this biopesticide [54]. In addition to these positive facts, a recent study demonstrated that *B. bassiana* Delta native NI8 and the commercial GHA (Grasshopper Isolate) (BotaniGard 22WP) strains affect the fecundity of *N. viridula* decreasing its fecundity 3.5-fold with NI8 and 5.4-fold with GHA when use concentrations of 7 × 10^7^ spores/mL. Moreover, same study reported that females are 15-fold more susceptible than males [43]. Overall, the strain NI8 is already stablished in the Mississippi Delta [13,42] and extensive laboratory host-range bioassays have demonstrated that neither NI8 nor GHA strains affect the most common beneficial insects present in cotton, soybean, and wild host plants of stink bugs including *N. viridula* [55].

## Figures and Tables

**Figure 1 jof-08-01164-f001:**
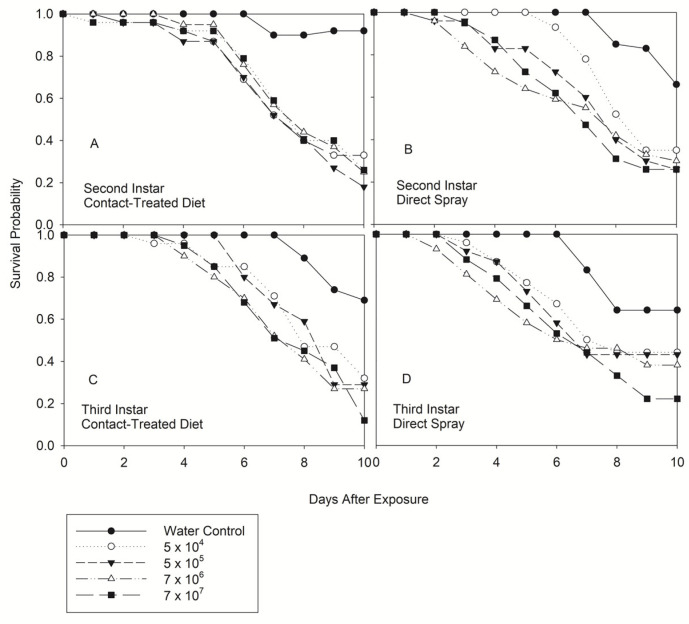
Product-limit survival estimates for 2nd and 3rd instar of *Nezara viridula* exposed by contact (treated diet) (**A**,**C**) and direct spray (**B**,**D**) to *Beauveria bassiana*, Delta native NI8 strain at different concentration. Insects were fed on an artificial diet (*p* = 0.05, LIFETEST of Equality over Strata).

**Figure 2 jof-08-01164-f002:**
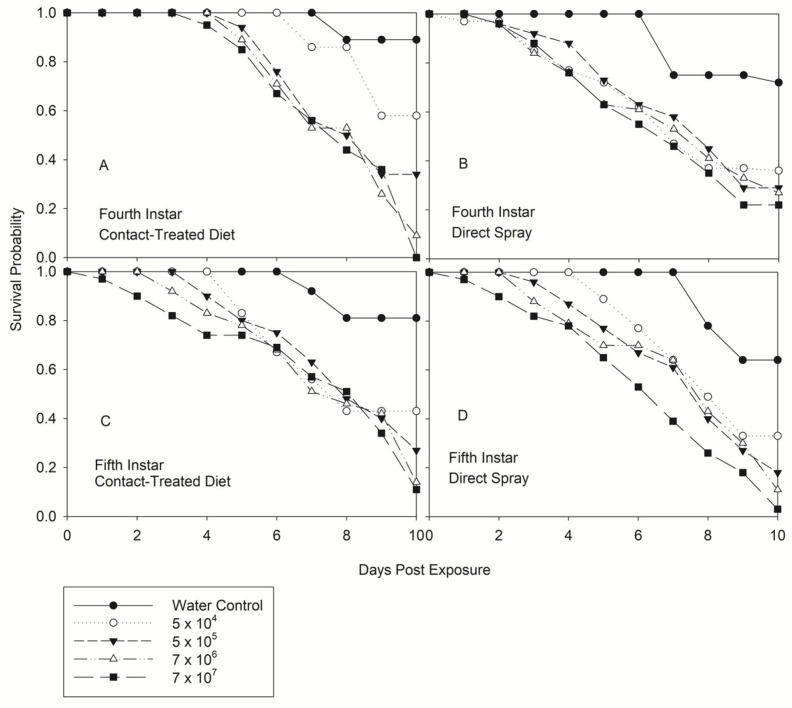
Product-limit survival estimates for 4th and 5th instars of *Nezara viridula* exposed by contact (treated diet) (**A**,**C**) and direct spray (**B**,**D**) to *Beauveria bassiana*, Delta native NI8 strain at different concentration. Insects were fed with an artificial diet (*p* = 0.05, LIFETEST of Equality over Strata).

**Figure 3 jof-08-01164-f003:**
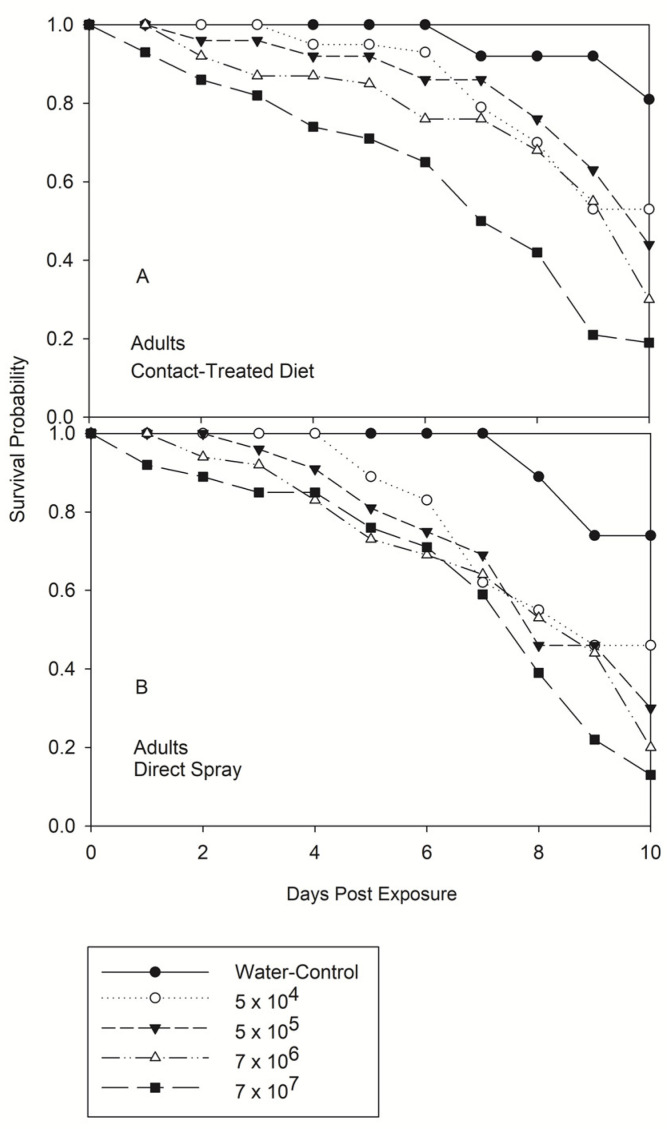
Product-limit survival estimates for adults of *Nezara viridula* exposed by contact (treated diet) (**A**) and direct spray (**B**) to *Beauveria bassiana*, Delta native NI8 strain at different concentrations. Insects were fed with an artificial diet (*p* = 0.05, LIFETEST of Equality over Strata).

**Figure 4 jof-08-01164-f004:**
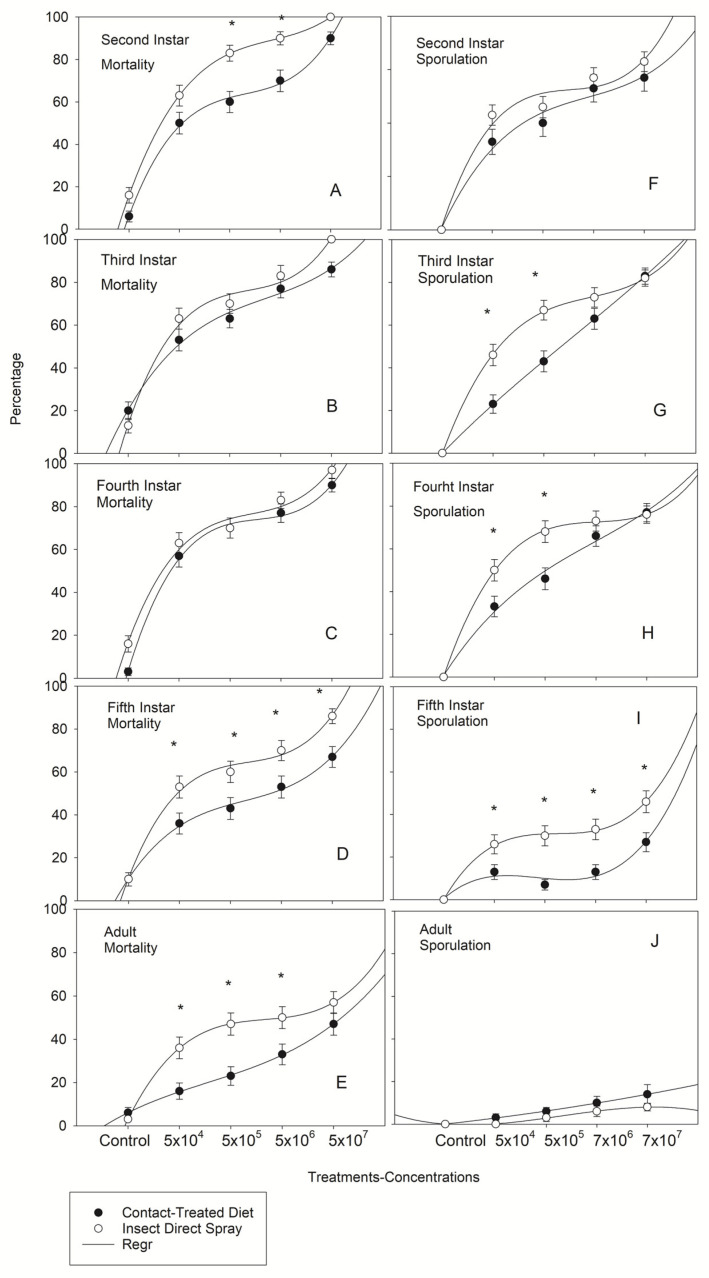
Regression (GLM) (Quadratic Trend Model) analysis predicting the probability of mortality percentage (**A**–**E**) of *Nezara viridula* nymphs (**A**–**D**) and adults (**E**) and sporulation percentage (**F**–**J**) of *Beauveria bassiana* as a function of dose level by contact and direct spray. (*) indicates that mortality and sporulation in Insect Direct Spray was significantly higher than Contact-Treated diet (Tukey’s HSD test, *p* = 0.05).

**Figure 5 jof-08-01164-f005:**
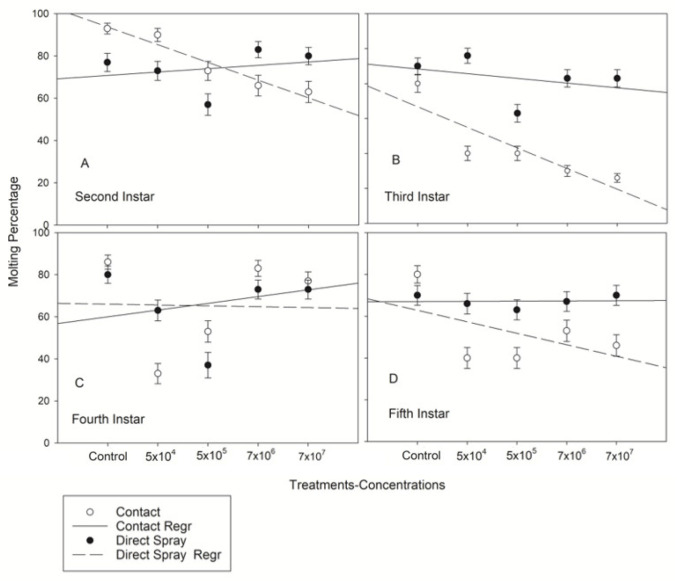
Regression (GLM) (Linear Trend Model) analysis predicting probability of molting percentage of *Nezara viridula* nymphs (**A**–**D**) as a function of dose level by contact and direct spray.

**Table 1 jof-08-01164-t001:** Test of Equality over Strata LIFETEST (Wilcoxon-Test) for nymphs and adults of *Nezara viridula* exposed by contact and direct spray to *Beauveria bassiana*, Delta native strain NI8 at different concentrations. Survival was scored for 10 d post-exposure.

Life Stages	Contact-Treated Diet			Direct Spray		
	X^2^	*DF*	*p* > X^2^	X^2^	*DF*	*p* > X^2^
2nd Instar	15.3653	4	0.0349	14.1056	4	0.0070
3rd Instar	9.8544	4	0.0430	10.6399	4	0.0309
4th Instar	11.9723	4	0.0176	8.9603	4	0.0621
5th Instar	10.6868	4	0.0303	16.6330	4	0.0023
Adult	11.9604	4	0.0176	8.6964	4	0.0692

Homogeneity Tests of survival curves (*p* = 0.05).

**Table 2 jof-08-01164-t002:** Lethal mortality-response (LC_50_) of different stages of *Nezara viridula* treated by contact (treated diet) with *Beauveria bassiana*, Mississippi Delta native strain NI8.

Stage	Concentration Response (Spores/mm^2^)							
	*n*	Slope ± SE	LC_50_ (95% CI)	Probit Trend				RR_50_ (95% CI) *
				Test for Slope		Test for GoF		
				X^2^	*p* > X^2^	X^2^	*p* > X^2^	
2nd Instar	150	0.43 ± 0.13	36.6 (6.3–70.9)	10.53	0.0012	0.52	0.8770	1.9 (0.37–9.30)
3rd Instar	150	0.43 ± 0.15	44.4 (3.4–98.9)	7.92	0.0049	0.91	0.5258	1.3 (0.18–9.14)
4th Instar	150	0.39 ± 0.13	17.8 (0.8–40.5)	9.17	0.0025	0.33	0.9747	1
5th Instar	150	0.32 ± 0.14	159.6 (38.3–2374)	5.22	0.0224	0.65	0.7680	6.4 (1.2–33.23)
Adult	150	0.41 ± 0.19	612.4 (241.5–4.4 × 10^6^)	4.72	0.0299	1.53	0.1211	30.5 (5.0–186.2)

Mortality was scored at 10 d post exposure. * Differences among RR_50_ values are significant if 95% CI do not include 1.0. RR_50_ compared the LC_50_s among the lowest LC_50_ as a control.

**Table 3 jof-08-01164-t003:** Lethal mortality-response (LC_50_) of different stages of *Nezara viridula* treated by direct spray with *Beauveria bassiana*, Mississippi Delta native strain NI8.

Stage	Concentration Response (Spores/mm^2^)							
	*n*	Slope ± SE	LC_50_ (95% CI)	Probit Trend				RR_50_ (95% CI) *
				Test for Slope		Test for GoF		
				X^2^	*p* > X^2^	X^2^	*p* > X^2^	
2nd Instar	150	0.66 ± 0.18	19.5 (3.8–36.6)	13.60	0.0002	0.84	0.5902	1.3 (0.2–8.4)
3rd Instar	150	0.46 ± 0.15	15.7 (0.6–36.8)	8.90	0.0029	1.17	0.3027	1
4th Instar	150	0.49 ± 0.20	22.8 (10.0–64.4)	5.71	0.0169	1.74	0.0668	1.3 (0.1–14.3)
5th Instar	150	0.32 ± 0.14	139.6 (28.8–1354)	5.22	0.0224	0.65	0.7680	10.0 (1.5–64.0)
Adult	150	0.19 ± 0.17	179.4 (-)	1.26	0.2612	1.95	0.0347	13.9 (1.1–175)

Mortality was scored at 10 d post exposure. * Differences among RR_50_ values are significant if 95% CI do not include 1.0. RR_50_ compared the LC_50_s among the lowest LC_50_ as a control.

**Table 4 jof-08-01164-t004:** Lethal Sporulation-response (LS_50_) of different stages of *Nezara viridula* treated by contact with *Beauveria bassiana*, Mississippi Delta native strain NI8.

Stage	Concentration Response (Spores/mm^2^)							
	*n*	Slope ± SE	LC_50_ (95% CI)	Probit Trend				RR_50_ (95% CI) *
				Test for Slope		Test for GoF		
				X^2^	*p* > X^2^	X^2^	*p* > X^2^	
2nd Instar	150	0.16 ± 0.12	197.3 (-)	1.87	0.1719	0.08	0.9999	3.8 (0.1–186)
3rd Instar	150	0.55 ± 0.16	61.7 (7.1–151)	12.40	0.0004	0.84	0.5863	1.2 (0.9–15.6)
4th Instar	150	0.31 ± 0.13	52.9 (0.0–328)	5.87	0.0154	0.68	0.7477	1
5th Instar	150	0.65 ± 0.22	869.0 (400–2 × 10^4^)	8.43	0.0037	0.73	0.7007	16.4 (1.2–222)
Adult	150	0.45 ± 0.64	1960 (-)	0.50	0.4816	0.75	0.6808	37.0 (0.1–9965)

Mortality was scored at 10 d post exposure. * Differences among RR_50_ values are significant if 95% CI do not include 1.0. RR_50_ compared the LC_50_s among the lowest LC_50_ as a control.

**Table 5 jof-08-01164-t005:** Lethal Sporulation-response (LS_50_) of different stages of *Nezara viridula* treated by direct spray with *Beauveria bassiana*, Mississippi Delta native strain NI8.

Stage	Concentration Response (Spores/mm^2^)							
	*n*	Slope ± SE	LC_50_ (95% CI)	Probit Trend				RR_50_ (95% CI) *
				Test for Slope		Test for GoF		
				X^2^	*p* > X^2^	X^2^	*p* > X^2^	
2nd Instar	150	0.15 ± 0.12	78.2 (-)	1.69	0.1939	0.42	0.9373	3.6 (0.03–407)
3rd Instar	150	0.36 ± 0.15	21.9 (6.6–185.8)	5.42	0.0198	1.21	0.2798	1
4th Instar	150	0.27 ± 0.12	26.2 (0.0–122.8)	4.87	0.0274	1.19	0.2894	1.2 (0.02–67.0)
5th Instar	150	0.22 ± 0.13	563.1 (-)	2.82	0.0934	0.69	0.7345	26 (04–1739)
Adult	150	0.11 ± 0.20	3.3 × 10^6^ (-)	0.28	0.5944	0.64	0.7845	1.5 × 10^6^ (5.5 × 10^–18^–3.4 × 10^31^)

Mortality was scored at 10 d post exposure. * Differences among RR_50_ values are significant if 95% CI do not include 1.0. RR_50_ compared the LC_50_s among the lowest LC_50_ as a control.

## Data Availability

Not applicable.

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
