# Peer review of "Lethal Concentration and Sporulation by Contact and Direct Spray of the Entomopathogenic Fungus Beauveria bassiana on Different Stages of Nezara viridula (Heteroptera: Pentatomidae)"

_jof, 2022, doi:10.3390/jof8111164_

Round 1

Reviewer 1 Report

The manuscript addresses an important topic of great consequence to agribusiness, public and environmental health, and to the control of agricultural pests in an eco-friendly way through biological control. The relevance of the topic addressed is a strong aspect of the work.

The work is well written, has an adequate introduction and clear objectives.

The methodology should be improved and better detailed in some points, especially in the evaluation of fungal sporulation. This is a point that deserves attention from the authors, since depending on the methodology used, the objective may not be achieved.

The results are well described.

The discussion is superficial and could be improved.

More details are described in the attached PDF.

Author Response

Please see all responses below

Line 14, change ‘lethal concentration’ to ‘median lethal concentration’

Response 1: Changed

Line 15, change ‘lethal sporulation’ to ‘median lethal sporulation’

Response 2: Changed

Line 19, change ‘Direct spray’ to ‘; Direct spray’

Response 3: Changed

Lines 20, 22, and 23, change ‘Spray’ to ‘; Direct spray’

Response 4: Changed

Lines 32-33, the sentence ‘Globally, is well known the low susceptibility of N. viridula to pyrethroids and organophosphates.’ is revised to ‘Globally, the low susceptibility of N. viridula to pyrethroids and organophosphates is well known.’

Response 5: Changed, now line 34.

Lines 43-44, change ‘often influence’ to ‘often influenced’

Response 6: Changed

Line 61, change ‘contributing’ to ‘contributed’

Response 7: Changed

Line 76, change ‘colony’ to ‘Colony’

Response 8: Changed

Line 77, change ‘has initially collected’ to ‘has been initially collected’

Response 9: Changed

Line 78, change ‘Delta. This colony’ to ‘Delta, which’

Response 10: No changes were done. If change to reviewer’s suggestion the sentence will be too long.

Line 80, delete ‘in’

Response 11: No changes. ‘in’ should not be deleted.

Line 84, change ‘isolate’ to ‘Isolate’

Response 12: Changed

Line 85, change ‘delta native NI8 strain’ to ‘Delta native strain NI8’

Response 13: Changed

Line 93, change ‘by contact: treated diet’ to ‘by Contact: Treated Diet’

Response 14: Changed

Line 95, change ‘70g’ to ‘70 g’; change ‘100g’ to ‘100 g’

Response 15: Changed

Line 96, change ‘150g’ to ‘150 g’

Response 16: Changed

Line 101, change ‘3000mL’ to ‘3000 mL’

Response 17: Changed

Line 106, change ‘direct spray)’ to ‘direct spray).’

Response 18: Changed

Lines 107 and 127, change ‘6mL’ to ‘6 mL’

Response 19: Changed

Line 109, change ‘24cm’ to ’24 cm’

Response 20: Changed

Line 110, change ’38.5cm’ to ’38.5 cm’

Response 21: Changed

Line 116, delete ‘by’

Response 22: Changed

Line 119, change ‘Same’ to ‘same’

Response 23: Changed

Line 124, change ‘the presence absence’ to ‘the presence or absence’

Response 24:

Line 126, change ‘direct spray’ to ‘Direct Spray’

Response 25: Changed

Line 129, change ‘rep’ to ‘replica’; change ‘, (experimental unit)’ to ‘(experimental unit),’

Response 26: It was changed to replication suggested by another reviewer

Line 144, please provide the full name of ‘ANOVA’

Response 27: Added

Line 155, change ‘Survival probability’ to ‘Survival Probability’

Response 28: Changed

Line 156, change ‘nymphal stage’ to ‘Nymphal Stage’

Response 29: Changed

In Table 1, change ‘Delta native NI8 strain’ to ‘Delta native strain NI8’

Response 30: Changed

Line 177, change ‘adults’ to ‘Adults’

Response 31: Changed

Line 186, change ‘Table1’ to ‘Table 1’

Response 32: Changed

Lines 191-192, change ‘sporulation percentage of Nezara viridula nymphs and adults among concentration’ to ‘Sporulation Percentage of Nezara viridula Nymphs and Adults among Concentrations’

Response 33: Changed

Line 208, change ‘direct spray. .’ to ‘direct spray.’

Response 34: Changed

Line 209, change ‘concentration’ to ‘concentrations’

Response 35: Changed

Lines 215-219, the sentence should be re-written.

Response 36: It is a title of the Fig. 4. Was not charged but added more information on the statistical analysis suggested by another reviewer.

Lines 223-224, change ‘sporulation response of Nezara viridula nymphs and adults to Beauveria bassiana strain NI8’ to ‘Sporulation Response of Nezara viridula Nymphs and Adults to Beauveria bassiana Strain NI8’

Response 37: Changed

Line 235, change ‘That lethal mortality’ to ‘Lethal mortality’

Response 38: Changed

Line 239, in Table 2, change ‘Mississippi Delta native NI8 strains’ to ‘Delta native strain NI8’

Response 39: Changed

Line 243, in Table 3, change ‘Mississippi Delta native NI8 strain’ to ‘Delta native strain NI8’

Response 40: Changed

Line 259, in Table 4, change ‘Mississippi Delta native NI8 strain’ to ‘Delta native strain NI8’

Response 41: Changed

Line 266, in Table 5, change ‘Mississippi Delta native NI8 strain’ to ‘Delta native strain NI8’

Response 42: Changed

Lines 270-271, change ‘percentage of Nezara viridula nymphs among concentration of Beauveria bassiana strain NI8’ to ‘Percentage of Nezara viridula Nymphs among Concentrations of Beauveria bassiana Strain NI8’

Response 43: Changed

Lines 273-274, change ‘concentration’ to ‘concentrations’

Response 44: Changed

Reviewer 2 Report

Line 14, change ‘lethal concentration’ to ‘median lethal concentration’

Line 15, change ‘lethal sporulation’ to ‘median lethal sporulation’

Line 19, change ‘Direct spray’ to ‘; Direct spray’

Lines 20, 22, and 23, change ‘Spray’ to ‘; Direct spray’

Lines 32-33, the sentence ‘Globally, is well known the low susceptibility of N. viridula to pyrethroids and organophosphates.’ is revised to ‘Globally, the low susceptibility of N. viridula to pyrethroids and organophosphates is well known.’

Lines 43-44, change ‘often influence’ to ‘often influenced’

Line 61, change ‘contributing’ to ‘contributed’

Line 76, change ‘colony’ to ‘Colony’

Line 77, change ‘has initially collected’ to ‘has been initially collected’

Line 78, change ‘Delta. This colony’ to ‘Delta, which’

Line 80, delete ‘in’

Line 84, change ‘isolate’ to ‘Isolate’

Line 85, change ‘delta native NI8 strain’ to ‘Delta native strain NI8’

Line 93, change ‘by contact: treated diet’ to ‘by Contact: Treated Diet’

Line 95, change ‘70g’ to ‘70 g’; change ‘100g’ to ‘100 g’

Line 96, change ‘150g’ to ‘150 g’

Line 101, change ‘3000mL’ to ‘3000 mL’

Line 106, change ‘direct spray)’ to ‘direct spray).’

Lines 107 and 127, change ‘6mL’ to ‘6 mL’

Line 109, change ‘24cm’ to ’24 cm’

Line 110, change ’38.5cm’ to ’38.5 cm’

Line 116, delete ‘by’

Line 119, change ‘Same’ to ‘same’

Line 124, change ‘the presence absence’ to ‘the presence or absence’

Line 126, change ‘direct spray’ to ‘Direct Spray’

Line 129, change ‘rep’ to ‘replica’; change ‘, (experimental unit)’ to ‘(experimental unit),’

Line 144, please provide the full name of ‘ANOVA’

Line 155, change ‘Survival probability’ to ‘Survival Probability’

Line 156, change ‘nymphal stage’ to ‘Nymphal Stage’

In Table 1, change ‘Delta native NI8 strain’ to ‘Delta native strain NI8’

Line 177, change ‘adults’ to ‘Adults’

Line 186, change ‘Table1’ to ‘Table 1’

Lines 191-192, change ‘sporulation percentage of Nezara viridula nymphs and adults among concentration’ to ‘Sporulation Percentage of Nezara viridula Nymphs and Adults among Concentrations’

Line 208, change ‘direct spray. .’ to ‘direct spray.’

Line 209, change ‘concentration’ to ‘concentrations’

Lines 215-219, the sentence should be re-written.

Lines 223-224, change ‘sporulation response of Nezara viridula nymphs and adults to Beauveria bassiana strain NI8’ to ‘Sporulation Response of Nezara viridula Nymphs and Adults to Beauveria bassiana Strain NI8’

Line 235, change ‘That lethal mortality’ to ‘Lethal mortality’

Line 239, in Table 2, change ‘Mississippi Delta native NI8 strains’ to ‘Delta native strain NI8’

Line 243, in Table 3, change ‘Mississippi Delta native NI8 strain’ to ‘Delta native strain NI8’

Line 259, in Table 4, change ‘Mississippi Delta native NI8 strain’ to ‘Delta native strain NI8’

Line 266, in Table 5, change ‘Mississippi Delta native NI8 strain’ to ‘Delta native strain NI8’

Lines 270-271, change ‘percentage of Nezara viridula nymphs among concentration of Beauveria bassiana strain NI8’ to ‘Percentage of Nezara viridula Nymphs among Concentrations of Beauveria bassiana Strain NI8’

Lines 273-274, change ‘concentration’ to ‘concentrations’

Line 282, change ‘direct spray. .’ to ‘direct spray.’

Line 289, change ‘IMP’ to ‘IPM’

Line 295, change ‘In our study’ to ‘In our study,’

Line 301, change ‘result’ to ‘results’

Line 316, delete ‘it’

Line 319, change ‘19 spores,’ to ‘19 spores /’

Line 323, change ‘concentration’ to ‘concentrations’

Line 327, change ‘indicated, that spraying’ to ‘indicated that spraying’

Line 329, change ‘give’ to ‘gives’

Line 353, delete ‘In’

The section of References should be checked thoroughly to confirm the correction for each reference.

Line 372, change ‘Insects.’ to ‘Insects’

Line 376, 442, 444, and 467, delete ‘of’

Line 399, please provide the abbreviation of the journal.

Line 408, change ‘Effectiveness of Fungus Beauveria bassiana on Mortality of Nezara viridula on Stadia Nymphs and Imago.’ to ‘Effectiveness of fungus Beauveria bassiana on mortality of Nezara viridula on stadia nymphs and imago.’

Line 418, change ‘Some Entomopatogen Fungus on Green Ladybug Imago (Nezara Viridula Linnaeus)’ to ‘some entomopatogen fungus on green ladybug imago (Nezara viridula Linnaeus)’

Line 430, change ‘Mycologya.’ to ‘Mycologia’

Line 437, change ‘Mycologia.’ to ‘Mycologia’

Line 439, please confirm the name of ‘Frontiers.’

Line 446, ‘Paecilomyces’ should be italicized; change ‘Infection’ to ‘infection’

Line 451, please provide the abbreviation of the journal.

Lines 452-454, change ‘Laboratory and Field Investigation on Compatibility of Beauveria bassiana (Hypocreales: Clavicipitaceae) Spores with a Sprayable Bioplastic Formulation for Application in the Biocontrol of Tarnished Plant Bug in Cotton’ to ‘Laboratory and field investigation on compatibility of Beauveria bassiana (Hypocreales: Clavicipitaceae) spores with a sprayable bioplastic formulation for application in the biocontrol of tarnished plant bug in cotton’

Line 465, change ‘Crop Protection’ to ‘Crop Prot.’

Line 475, change ‘Entomopathogenic’ to ‘entomopathogenic’

Line 476, ‘Nezara viridula’ should be italicized.

Lines 477-478, change ‘Lethality of the Entomogenous Fungus Beauveria bassiana Strain NI8 on Lygus lineolaris (Hemiptera: Miridae) and its Possible Impact on Beneficial Arthropods’ to ‘Lethality of the entomogenous fungus Beauveria bassiana strain NI8 on Lygus lineolaris (Hemiptera: Miridae) and its possible impact on beneficial arthropods’; ‘Lygus lineolaris’ should be italicized.

Reviewer 3 Report

This is a well done study and the results are very well presented. Since I have no suggestions for improvement, I recommend the study for publication.

Author Response

No comments from Reviewer # 3